# Amundsen Sea circulation controls bottom upwelling and Antarctic Pine Island and Thwaites ice shelf melting

Taewook Park[1] ✉, Yoshihiro Nakayama [2] ✉ & SungHyun Nam [3]

The Pine Island and Thwaites Ice Shelves (PIIS/TIS) in the Amundsen Sea are melting rapidly and impacting global sea levels. The thermocline depth (TD) variability, the interface between cold Winter Water and warm modified Circumpolar Deep Water (mCDW), at the PIIS/TIS front strongly correlates with basal melt rates, but the drivers of its interannual variability remain uncertain. Here, using an ocean model, we propose that the strength of the eastern Amundsen Sea on-shelf circulation primarily controls TD variability and consequent PIIS/TIS melt rates. The TD variability occurs because the on-shelf circulation meanders following the submarine glacial trough, creating vertical velocity through bottom Ekman dynamics. We suggest that a strong or weak ocean circulation, possibly linked to remote winds in the Bellingshausen Sea, generates corresponding changes in bottom Ekman convergence, which modulates mCDW upwelling and TD variability. We show that interannual variability of off-shelf zonal winds has a minor effect on ocean heat intrusion into PIIS/TIS cavities, contrary to the widely accepted concept.

Ice shelves in the vicinity of Pine Island Bay (PIB) have experienced significant melting in recent decades[1,2], contributing to sea level rise[3,4]. Recent estimates suggest that, from 2012 to 2017, the melting of Pine Island and Thwaites Glaciers has contributed over 0.25 mm annually to global sea-level rise[5,6]. Prior studies[4,7–9] have predicted that ice shelf melting and grounding line retreat will accelerate. Because it is grounded below sea level, Thwaites Glacier is believed to be susceptible to irreversible loss of mass with continued rapid acceleration of the ice flow[10] and retreat of the ice front[11,12] and grounding line[13]. Substantial retreat of the grounding line raises the possibility of a collapse of Thwaites Glacier, which would account for more than half a metre of future global sea-level rise and destabilise neighbouring glaciers, representing a further 3 m of global sea-level rise[14].

Basal melting of Pine Island Ice Shelf and Thwaites Ice Shelf (PIIS/TIS) is primarily caused by heat delivery into the sub-ice shelf cavities by the warm modified Circumpolar Deep Water (mCDW)[15–18]. The mCDW crosses the continental shelf break through two main troughs[17,19,20], travels a few hundred kilometres on the continental shelf, and then enters the sub-ice shelf cavities[21–23], thereby contributing to basal melting. The basal melt rates exhibit interannual variability, as revealed by extensive observations over the past 15 years[16,21] and with a focus on the years 2009–2014[24]. They are controlled by the amount of mCDW inflow and/or variation in the thermocline depth (TD)[15,25,26], where the TD, characterised by a sharp temperature rise with depth, is the interface between cold Winter Water and warm mCDW. The bottom geometry in the PIB, especially the ridge located beneath the PIIS between the inner and outer cavities (red arrow in Fig. 1c), is suggested to play a crucial role in regulating ocean circulation into and out of the sub-ice shelf cavities and controlling ice shelf melting and geometry because (1) the TD determines the shallowest depth of mCDW and (2) bathymetric elevation prevents the densest and warmest mCDW from flowing into sub-PIIS cavities[16,21,27]. Changes in ocean circulation in front of the cavities or underneath the ice shelves[16,23,28,29] and cyclonic and anticyclonic gyres near the ice shelves[29] have been reported to have a significant effect on basal melting.

[1]Division of Ocean and Atmosphere Sciences, Korea Polar Research Institute, Incheon 21990, Republic of Korea. [2]Institute of Low Temperature Science, Hokkaido University, Sapporo 060-0819, Japan. [3]School of Earth and Environmental Sciences/Research Institute of Oceanography, Seoul National University, Gwanak-gu, Seoul 08826, Republic of Korea. ✉e-mail: twpark@kopri.re.kr; Yoshihiro.Nakayama@lowtem.hokudai.ac.jp

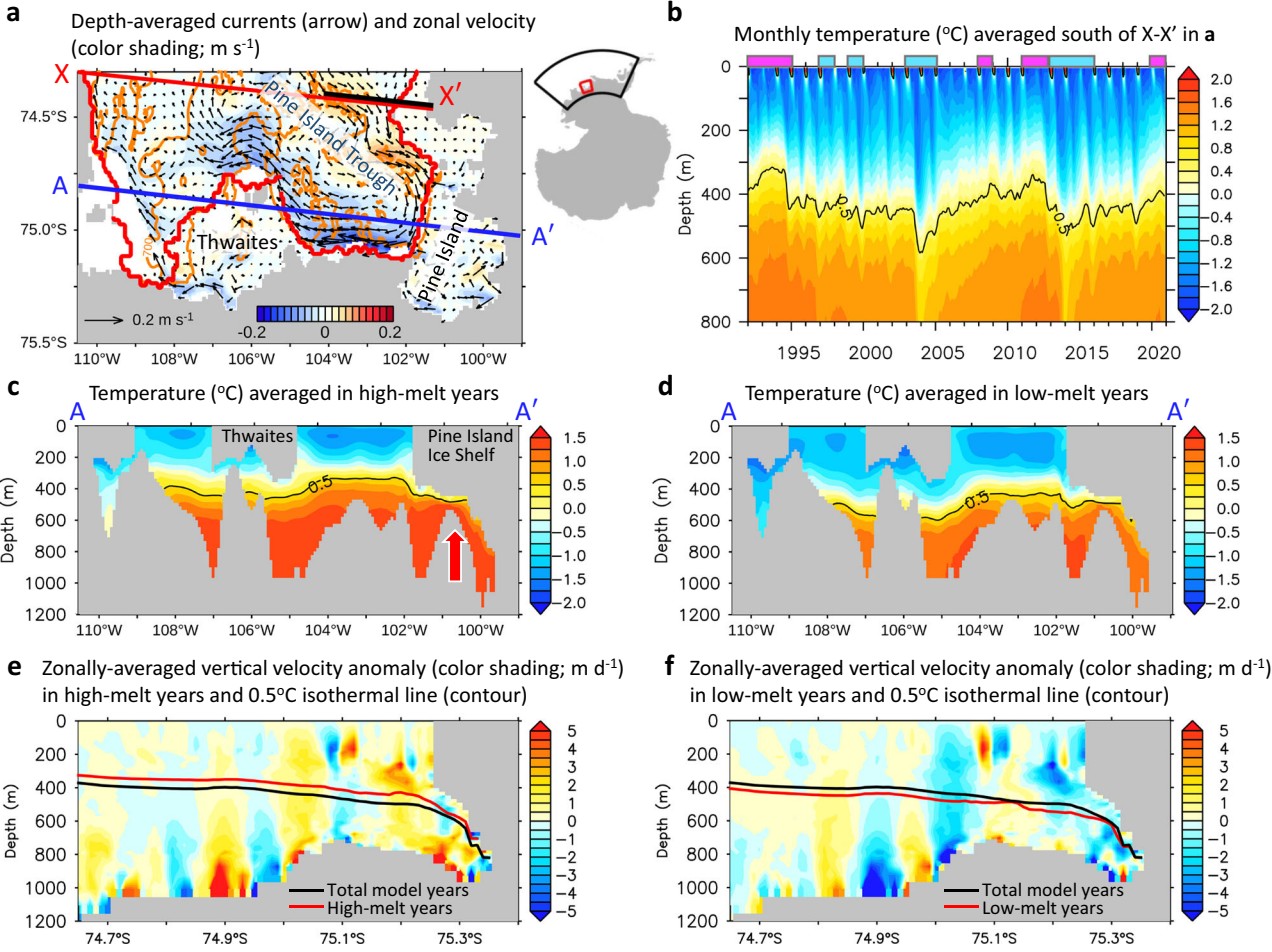

**Fig. 1 | Thermocline depth changes, and comparisons of temperature and vertical velocities in high- and low-melt years. a** Depth-averaged ocean currents (arrows) and zonal velocity (colour shading; m s⁻¹). A yellow colour-coded contour indicates the bathymetry of 700 m depth. The blue line (A–A′) denote the vertical sections corresponding to the plots in **c** and **d**. The thick black line represents the section defined to calculate the Antarctic Coastal Current. In the inset map, the black line represents the model domain and the red box represents the study area. **b** Depth-time plot of monthly-averaged temperature (°C) averaged south of X–X′ in **a**. Magenta and cyan colour bars at the top of the graph indicate high-melt years (1992, 1993, 1994, 2008, 2011, 2012, 2020) and low-melt years (1997, 1999, 2003,

2004, 2013, 2014, 2015), respectively (Methods). **c**, **d** Composite average temperature (°C) across section (A–A′ in **a**) averaged in high- and low-melt years, respectively. The black contour line in each figure depicts 0.5 °C isothermal line averaged in high- and low-melt years. The red colour-coded arrow indicates the ridge between inner and outer cavity of Pine Island Ice Shelf. **e**, **f** Composite anomaly of vertical velocity (m d⁻¹), indicating the deviation from the total mean of the model simulation, averaged zonally within the domain in **a** for high-melt and low-melt years, respectively. The black contour line represents 0.5 °C isothermal line averaged for the total model simulation, while the red contour line shows 0.5 °C isothermal line averaged in high- and low-melt years in each figure.

Several mechanisms have been proposed to explain the variability in the mCDW inflow and the TD in the PIB. First, westerly winds can generate dense mCDW moving up onto the shelf and towards the main trough, leading to the accumulation of mCDW in the PIB[15,21,30,31]; El Niño Southern Oscillation (ENSO)-related westerly winds over the shelf break[21,32–34] are hypothesised to be linked to the on-shelf flow of mCDW. A strong link between ENSO and ice shelf melting are examined focusing on the variations in both ice shelf thickness and mass[35]. Several studies[19,20,26] have suggested that undercurrents along the outer continental shelf break encounter troughs and turn onshore, causing mCDW onshore intrusion. The undercurrent is suggested to weaken when easterly anomaly winds develop at the shelf break[26,36], while the relationship is opposite at a decadal timescale[37]. Second, the wind-driven Ekman pumping in the continental shelf and at the shelf break are suggested to promote TD fluctuations and enhance onshore volume transport[26,38]. Third, sea ice formation and cooling due to local air-sea heat exchange within the polynyas in the PIB can drive interannual variability in TD[24,39]. However, prior modelling studies have not been carried out specifically aiming to reproduce and explain the TD variability in the PIB.

Here we hypothesize that TD variability is closely linked with the strength of ocean circulation and bottom Ekman dynamics and use the MITgcm regional Amundsen-Bellingshausen Sea simulation[40] (Methods), which successfully simulates TD variability in good agreement with observations, to investigate its main drivers.

## Results

### Thermocline depth variability governing melt rates

We investigated the ocean circulation and TD variability in the PIB using the model simulation from 1992 to 2020. The pattern of the full depth-averaged ocean currents (Fig. 1a) revealed an inflow from the north entering the PIB through the eastern side of the Pine Island Trough and flowing out of the PIB from the western side, forming a cyclonic circulation. The ocean circulation transports warm mCDW (about 0.5 °C or warmer located below 400–600 m depth) into the cavities of the PIIS/TIS and towards the grounding lines (Fig. 1c, d). A depth-time plot of the temperature in the PIB reveals that the TD, which defines 0.5 °C isotherm depth separating upper Winter Water and lower mCDW, has multi-timescale variability (Fig. 1b). The thickness and temperature of the lower warm water layer, which

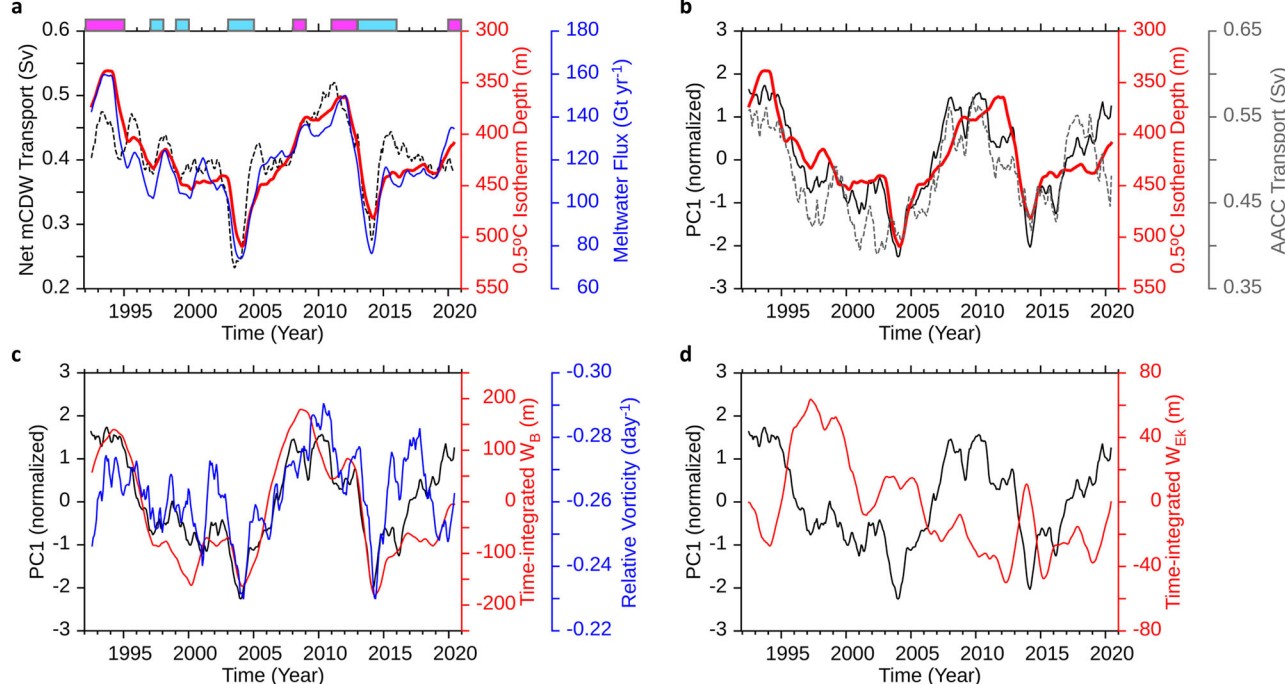

**Fig. 2 | Thermocline depth and meltwater flux, and their causing factors. a** Net modified Circumpolar Deep Water (mCDW) transport (black dotted line) across the boundary (X–X') in Fig. 1a, 0.5°C isotherm depth (red line) averaged south of X–X' in Fig. 1a, and meltwater flux (blue line) from Thwaites and Pine Island Glaciers. Magenta and cyan colour bars at the top of the graph indicate high- and low-melt years, respectively (Methods). **b** The leading mode's principal component (PC1) of ocean current (black line) (Methods), southward Antarctic Coastal Current (AACC) transport (grey dotted line) across 74.4°S, 104–101.5°W in X–X' in Fig. 1a, and 0.5 °C isotherm depth (red line) as in **a**. **c** The PC1 (black line) as in **b**, depth-averaged relative vorticity (blue line) averaged south of X–X' in Fig. 1a, and time-integrated vertical velocity ($W_B$) anomaly in the bottom layer (red line) (Methods). **d** The PC1 (black line) as in **b** and time-integrated Ekman pumping velocity ($W_{Ek}$) anomaly with positive values indicating upwelling, derived by ocean surface stress curl (red line) (Methods). Both the time-integrated vertical velocity, denoted as $W_{Ek}$ in **c** and $W_B$ in **d**, were averaged south of A–A' in Fig. 1a to compare their relative magnitudes. Seasonal variability in all the quantities in **a–d** is removed by applying a 13-month moving average filter.

plays a crucial role in ice shelf melting, exhibits interannual to decadal variability, with a maximum vertical displacement of ~300 m from 1992 to 2020. This corresponds to isotherm fluctuations with similar amplitude observed over 5 years in a previous observational study[24].

Our results demonstrate a strong correlation between the TD in the PIB and basal melt rates of PIIS/TIS (Supplementary Fig. 1), exhibiting year-to-year variation (red and blue lines in Fig. 2a). Low-melt years, such as 2004 and 2014, showed a deeper TD compared to other years, while the TD is shallower in high-melt years, such as 1993 and 2012 (Fig. 2a). Notably, the TD is strongly correlated with the volume transport of mCDW inflow into the PIB; as the mCDW inflow increases, the thermocline lifts, resulting in higher melt rates, and vice versa in years of lower melt rates. We calculated composites of high- and low-melt years ("Methods" section) and found that during high-melt years, the thermocline was uplifted and positioned close to the bottom of the ice shelf (Fig. 1c, d). This displacement means that more heat is delivered into the sub-ice shelf cavities, which can increase the basal melt rates. Our findings are consistent with previous observational and modelling studies[16,23,38,41], particularly observations from an underwater vehicle[42], which highlight the significant impact of the vertical displacement of the TD on changes in PIIS/TIS basal melt rates.

### Ocean circulation controlling thermocline depth

Empirical Orthogonal Function (EOF) analysis was conducted to identify the primary mode of ocean circulation (Methods). The results of the leading EOF mode indicated that the current flowed onshore along the eastern side of the Pine Island Trough, approaching the ice shelves, and then flowed offshore through the western side (Fig. 3a).

The temporal variability of this structure was captured using a Principal Component (PC) timeseries, as depicted in Fig. 3b, indicating that high and low PC values correspond to stronger and weaker cyclonic ocean circulation in the EOF spatial patterns, respectively.

Our analysis revealed a significant relationship between ocean circulation and TD changes, with a confidence level of 95% (Fig. 2b, black and red lines). For example, when the cyclonic ocean circulation weakened in 2004 and 2014, the thermocline deepened, whereas in 1993 and 2010, and 2020, when the cyclonic ocean circulation was stronger, the TD was lifted. This suggests that cyclonic ocean circulation in the PIB connected to the eastern Amundsen Sea is a primary cause of the TD displacement in the PIB. Moreover, we note that this ocean circulation coincides with the southward transport of the coastal current in the PIB, a larger-scale ocean current system (Fig. 2b, black and grey dotted lines), providing further evidence that the southward-flowing current in the eastern Amundsen Sea is closely connected to on-shelf circulation including the Antarctic Coastal Current (AACC)[43].

To explain how ocean circulation modulates TD displacement, we suggest that the horizontal convergence at the bottom Ekman layer is responsible for the upwelling of warm mCDW into the ocean interior, which, in turn, affects the displacement of the TD. This hypothesis is supported by the result that time-integrated vertical velocity fluctuates similarly to the TD (Fig. 2c). This analysis suggests that stronger cyclonic ocean circulation may enhance mCDW convergence, as indicated by the larger negative relative vorticity (Methods) (Fig. 2c). This relationship is further supported by the composite maps (Fig. 1e, f), which demonstrate that the upward velocity is particularly strong at greater depths in high-melt years than in low-melt years. We revisited our analysis by using monthly data to investigate the seasonal

**a**

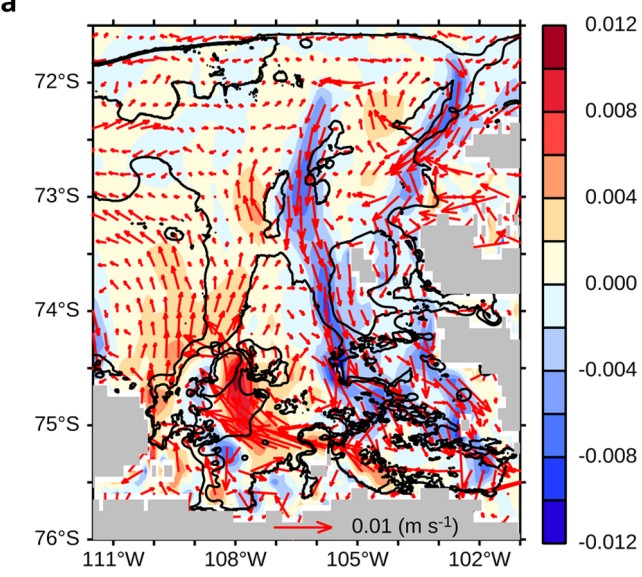

**b**

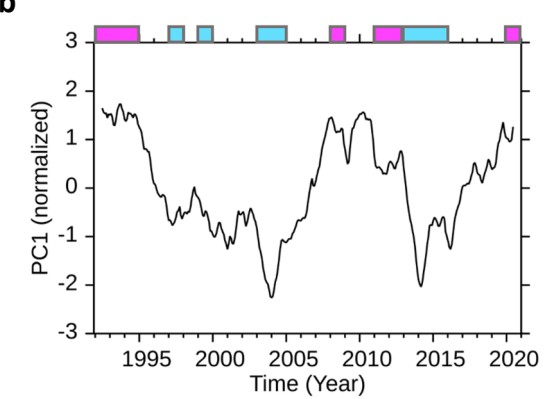

**Fig. 3 | Primary pattern of ocean circulation and corresponding principal component timeseries. a** The leading Empirical Orthogonal Function (EOF) mode (m s⁻¹) of depth-averaged ocean current, explaining about 33% of the total variance. Meridional velocity of the EOF is shaded in colour. **b** The corresponding principal component (PC1) timeseries (normalised). Magenta and cyan colour bars at the top of the graph indicate high-melt years (1992, 1993, 1994, 2008, 2011, 2012, 2020) and low-melt years (1997, 1999, 2003, 2004, 2013, 2014, 2015), respectively. A 13-month moving average filter is applied prior to the EOF analysis.

variability and compared this with the identified interannual variability (Supplementary Fig. 2a, b). We found that the pronounced seasonal variability in ocean currents does not manifest similarly in the TD because response of the TD is moderated by the slower and cumulative changes in vertical velocity in the bottom layer, which filters out short-term fluctuations, such as those that occur during seasonal periods (Supplementary Fig. 2c).

We performed an analysis of the depth-integrated vorticity budget that takes into account the time tendency of vorticity, relative vorticity advection, along with the surface and bottom vortex stretching terms (Methods) (Supplementary Fig. 3). The most prominent factors were the advection of relative vorticity and the bottom vortex stretching term. We noted that the tendency of vorticity is negligibly small, which can become important in shorter timescales, e.g., super-inertial scales such as daily or shorter, but not over monthly or longer timescales relevant to this study. The residual term, which includes effects of mixing and computational errors, contributes to a minor 3% of the relative vorticity advection. Notably, the bottom vortex stretching term primarily acts in response to the advection of relative vorticity. Overall, our results suggest that enhanced vertical velocity due to the

strengthened on-shelf circulation along bathymetric troughs in the PIB contributes to thermocline lifting through the amplification of cyclonic oceanic conditions.

We investigated the influence of surface Ekman pumping, generated by instantaneous ocean surface stress curl, on the TD variability using a similar approach to previous studies[26,38]. The surface Ekman pumping does not exhibit coherent variations with the TD variability; occasionally, it presents an opposing phase to the TD variability (Supplementary Fig. 4). Additionally, the magnitude of the TD variability estimated from the time-integrated surface Ekman pumping (Fig. 2d) is ~70% smaller than that estimated from time-integrated near-bottom vertical velocity (Fig. 2c). This underlines the minor role of surface Ekman pumping in influencing TD variability, which is consistent with results of previous studies[33,37]. These studies have shown that surface Ekman pumping is unlikely to be the main driver of influencing the variability in CDW layer thickness in the eastern Amundsen Sea on interannual or decadal timescale. A number of prior studies have investigated the influence of buoyancy forcing on the melting of ice shelves. Several studies have proposed that sea ice formation and cooling due to local air-sea heat exchange within the polynyas in the PIB can drive seasonal fluctuations in the TD[24,39]. Meanwhile, it is suggested that the local surface buoyancy forcing is unlikely to drive the variability of the ice shelf melting in the eastern Amundsen Sea on a decadal timescale[37]. While we mainly examined the variability on interannual to decadal timescales, we cannot rule out that surface buoyancy forcing could also affect variability shorter than those timescales.

### Roles of ocean surface stress on ocean circulation

We investigated the contribution of Ocean Surface Stress (OSS) to the variability in ocean circulation. The OSS was computed from wind forcing and modified in the ocean model by the motion and concentration of sea ice. Prior studies[21,36,37] have hypothesised that westerly winds off the continental shelf break favour onshore mCDW intrusion, thereby promoting more melting conditions on the PIB. Conversely, in our study, local and remote wind forcings over the continental shelf are proposed to be linked with increased ice shelf melting (Fig. 4a, d).

Locally in the eastern Amundsen Sea, our investigation on the OSS found a moderate correlation between the northward OSS and the enhanced cyclonic ocean circulation at a 90% confidence interval (Methods) (Fig. 4d, f). Remotely, we found that the easterly winds in the Bellingshausen Sea and West Antarctic Peninsula continental shelves are moderately correlated with on-shelf circulation in the eastern Amundsen Sea at a 90% confidence level (Fig. 4a), indicating that remote easterly winds can enhance the AACC, thus strengthening ocean circulation downstream in the PIB. This is consistent with previous studies showing that the prevailing easterly winds were examined as a driving force for AACC variability, along with buoyancy forces from meltwater and runoff in the Bellingshausen Sea and the West Antarctic Peninsula continental shelves[44–48]. Although a time lag may exist on a monthly scale, our analysis with 13-month moving averages suggests that the response is near instantaneous and we did not reveal any time lag between the winds in the Bellingshausen Sea and the ocean circulation in the eastern Amundsen Sea (Supplementary Fig. 5).

We investigated the evolution of the AACC between the Bellingshausen and Amundsen Seas and identified a strong and consistent connection between the on-shelf circulation in the eastern Amundsen Sea and various locations along the path of the AACC (Supplementary Fig. 6). This finding confirms that the AACC influences the ocean circulation in the eastern Amundsen Sea downstream instantaneously, mainly through its westward transport pathway[46,47]. The West Antarctic continental shelf is characterized by strong circumpolar winds that encircle the continent. These winds can generate coastal-trapped Kelvin waves that rapidly transmit barotropic current anomalies westward along the Antarctic coast

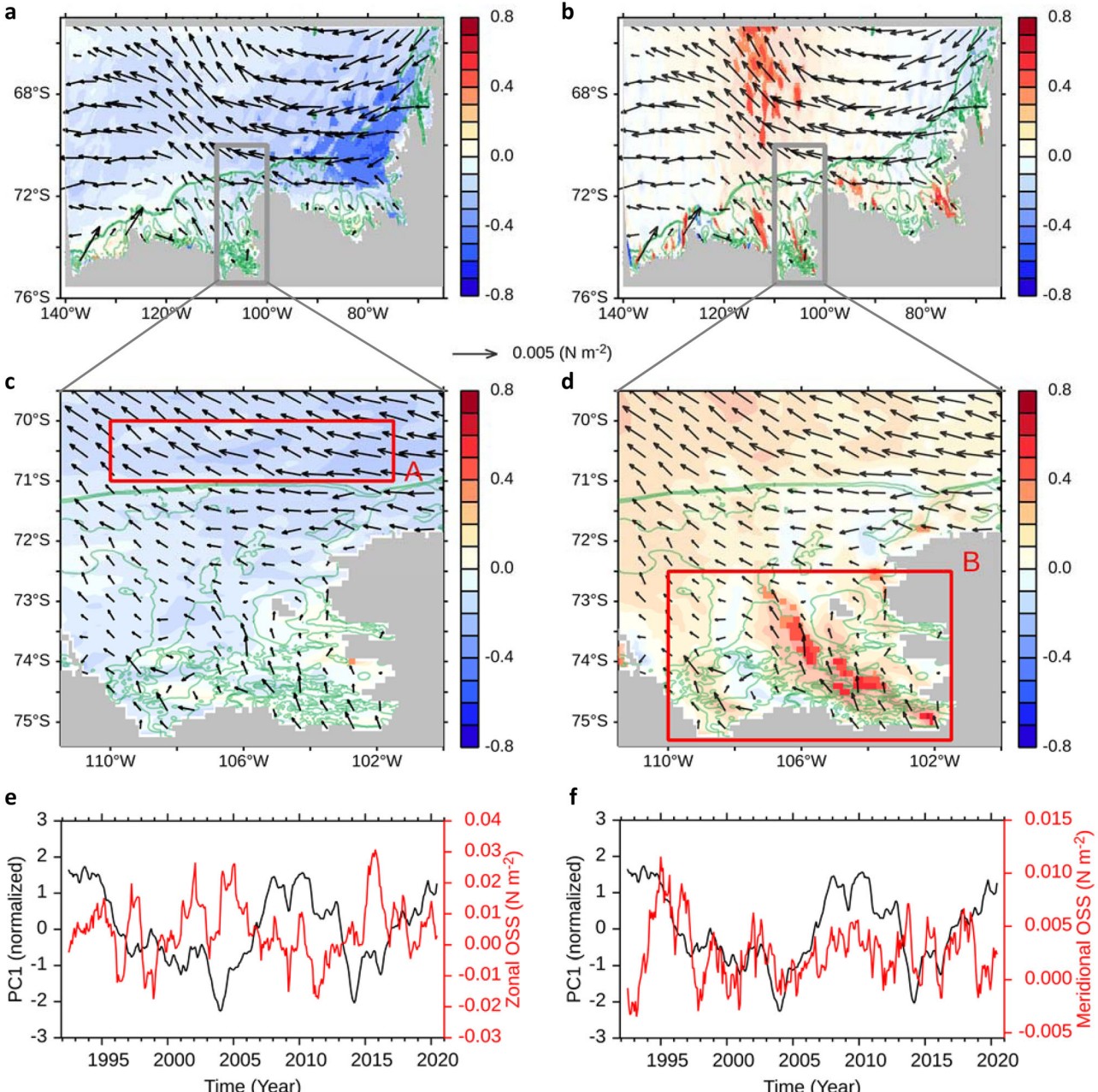

**Fig. 4 | Relationship between ocean currents and ocean surface stress.**
**a**, **b** Regression coefficients (arrows) of Ocean Surface Stress (OSS) anomalies on the leading mode's principal component (PC1) of ocean current, and the correlation coefficients (colour shading) between the PC1 and both zonal and meridional OSS, in **a** and **b**, respectively. Statistically insignificant regions in the 90%-level are transparently shaded. A green colour-coded contour indicates the bathymetry of 500 m and 700 m depth. **c**, **d** The same as in **a**, **b**, but zoomed on the grey box region in **a**, **b**. **e**, **f** The PC1 of ocean current (black line) in comparison with the zonal OSS (red line) averaged off the continental shelf break (region A in **c**) and meridional OSS (red line) averaged on the shelf (region B in **d**), respectively.

within hours to days[49,50]. This atmospheric connectivity can lead to simultaneous weather patterns across these regions, contributing to the near-zero lag in conditions.

To verify agreement with previous studies[21,36,37] on the roles of zonal winds, we examined whether the westerly winds at the shelf break significantly affect ice shelf melting on the PIB. The response of ocean circulation to wind forcing can be immediate or with some time delay. To investigate immediate ocean response to winds, we computed a regression and correlation map (Fig. 4c) between the OSS and cyclonic ocean circulation (Methods). Our results indicate that westerly winds off the continental shelf break are not significantly related to the strength of the on-shelf cyclonic ocean circulation. The correlation coefficients between the zonal OSS

averaged off the continental shelf break (box A in Fig. 4c) and the strength of the cyclonic ocean circulation resulted in only −0.22, indicating a statistically negligible correlation (Fig. 4e). We assessed the time-integrated ocean surface stress at the shelf break using an approach similar to that of previous studies[21,36] and compared it with the ocean circulation on the shelf by PC timeseries (Supplementary Fig. 7a). The PC response, even after accounting for time-integration of the winds, does not coincide directly with the wind forcing; rather, the winds precede by 2 years the ocean circulation. To investigate decadal ocean response to winds, we further conducted our analysis using a 5-year moving average filter (Supplementary Fig. 7b), consistent with a prior study[37]. We found a generally negative correlation at a decadal timescale between the westerly winds and on-shelf

ocean circulation, aligning with a previous study[37] that linked an easterly wind anomaly to an eastward undercurrent that can promote the heat transport onto the shelf. Nonetheless, our model output is not long enough to evaluate the robustness of this mechanism.

## Discussion

We propose that the variability in TD, and thus ice shelf melting, is controlled by the on-shelf circulation in the eastern Amundsen Sea and its pathway following the complex Antarctic coastline through bottom Ekman dynamics. The on-shelf circulation meanders along the coastline and forms a clockwise circulation near the PIIS/TIS, which induces upwelling. We advance our understanding of the role of winds, indicating that the southerly local winds in the PIB, along with the remote easterly winds over the continental shelf in the Bellingshausen Sea, are crucial factors in strengthening the circulation in the eastern Amundsen Sea. We did not find a significant relationship between westerly winds off the continental shelf break and ice shelf melt rates within the PIB on the interannual timescale, despite prior studies[21,36] suggesting that zonal winds at the shelf break could influence ice shelf melting.

Given that the on-shelf circulation following submarine glacial troughs is a widespread feature of the entire Antarctic coast and that the coastline has similar meandering structures due to a long history of glacier erosion, we infer that the proposed mechanism could regulate TD in other regions around Antarctica. Our findings provide important insights into the driving mechanisms of basal melting of the two ice shelves in the eastern Amundsen Sea and have implications for predicting future sea-level rise. Further studies on the upstream influences of the AACC and the wind forcings in the Bellingshausen Sea can help to elucidate the mechanism for the melting of Antarctic ice shelves. Our study underscores the need for long-term monitoring and numerical modelling, particularly focusing on the ocean circulation close to rapidly melting ice shelves, to better predict future Antarctic ice loss in this critical region.

## Methods

### MITgcm model and validation

We use a regional Amundsen Sea (AS)-Bellingshausen Sea (BS) configuration of the Massachusetts Institute of Technology general circulation model (MITgcm), which includes dynamic/thermodynamic sea ice and thermodynamic ice shelf capabilities. We use the model output[51] extended from 1992 to 2020, based on the AS-BS simulation from the study[40]. The model domain contains the AS and BS (black line in Fig. 1 inset) and has a nominal horizontal grid spacing of 1/12°, equivalent to 2–3 km over the AS and BS continental shelf. The vertical discretisation comprises 70 levels varying in thickness from 10 m near the surface, to 70–90 m at depths of 500–1000 m, and 450 m at the deepest level of 6000 m. The model bathymetry is the International Bathymetric Chart of the Southern Ocean (IBCSO)[52], and the model ice draft is Antarctic Bedrock Mapping (BEDMAP-2)[53]. The initial conditions are derived from a 16-year (2001–2016) spin-up, integrated from rest and from January World Ocean Atlas 2009 temperature[54] and salinity[55] fields. Surface and lateral forcing for the 1992–2017 period is provided by ECCO LLC270[56], which is based on ERA-Interim[57] and has been adjusted using the ECCO adjoint model-based methodology[58].

This model and its further downscaled version have been used and evaluated for off-shelf and on-shelf hydrography and circulation in the Amundsen Sea and Bellingshausen Sea[23,40,59–61]. We compared model results with observations from a prior study (see Fig. 2c in Webber et al.[24].), which obtained timeseries of ocean temperature in the vicinity of the ice front of the Pine Island Ice shelf on a monthly basis from 2009 to 2014. During the same period at the nearest model grid point to the observation site, we demonstrated that TD varies with

a range of approximately 290 m for 5 years, from 620 m to 330 m, which is similar to the observational range of approximately 300 m for 5 years, but with maximum and minimum depths of 700 m and 400 m, respectively, indicating our model underestimated the TD during this period. While the observational data indicates that the thermocline deepens from 2011 reaches its maximum depth in early 2013, our model indicates the thermocline deepens from 2012 with the deepest TD in October 2013, with an acceptable time delay for our study, which focuses on a longer timeseries than seasonal variation. Despite the discrepancy between model and observation for the time-averaged TD, we believe that our model accurately represents TD variability.

### Definition of high- and low-melt years

Based on one standard deviation of the simulated monthly melt rates of both Pine Island and Thwaites Ice Shelves, high-melting and low-melting cases were defined for composite analysis of TD and vertical velocity. High-melting cases were selected from 1992, 1993, 1994, 2008, 2011, 2012, and 2020, and low-melting cases were selected from 1997, 1999, 2003, 2004, 2013, 2014, and 2015.

### Empirical Orthogonal Function analysis

EOF analyses are widely used to quantify the spatial and temporal variability of a variable of interest. In the present study, we investigated the leading modes of ocean circulation by combining meridional and zonal fields. Prior to EOF analysis, seasonal variability was eliminated using a 13-month moving average, and a geographic weighting factor was applied based on latitude. The use of a 13-month moving average filter serves to remove the effects of shorter-term fluctuations, such as those associated with seasonal variability. Consequently, the correlations identify relationships over interannual to decadal timescales. As a result of the EOF analysis, the leading mode was used to represent major ocean circulation and temporal variability, known as the Principal Component (PC) timeseries. The PC timeseries were normalised by their respective standard deviations. We abbreviate the leading mode of the PC timeseries as PC1.

### Regression and correlation analysis

Regression and correlation analyses were used to determine the relationships between two variables. Linear regression coefficients between ocean surface stress (zonal and meridional components) and the EOF PC timeseries of ocean currents indicated changes in ocean surface stress per unit change in the PC timeseries of ocean currents. Arrows indicate the regression coefficients for the zonal and meridional stresses. Correlation coefficients were calculated to measure the strength of the linear relationship between ocean surface stress and the PC timeseries. Student's $t$ test was conducted with lag-one autocorrelation coefficients for the effective sample size to test the significance of the coefficient[62].

### Relative vorticity, surface Ekman pumping velocity, and vertical velocity at the bottom

Relative vorticity ($\zeta$) is defined as

$$\zeta = \nabla \times \mathbf{u} \tag{1}$$

where $\mathbf{u}$ is the horizontal velocity of the ocean current. The depth-averaged relative vorticity is used for Fig. 2c in the main text. The Ekman pumping velocity ($w_{Ek}$) derived from ocean surface stress is calculated as

$$w_{Ek} = \frac{1}{\rho_0 f_0} \nabla \times \boldsymbol{\tau_o} \tag{2}$$

where $\boldsymbol{\tau_o}$ is horizontal ocean surface stress, $\rho_o = 1030 \, \text{kg m}^{-3}$ is ocean density, and $f_o = 2\Omega \sin\theta$ is the Coriolis parameter at the latitude $\theta$

where the southern hemisphere has a negative sign, $\Omega = 7.29 \times 10^{-5}\,\mathrm{s}^{-1}$ is the angular speed of rotation of the Earth. A positive (negative) value of $w_{\mathrm{Ek}}$ denotes upwelling (downwelling) motion.

The vertical velocity near the bottom ($w_{\mathrm{B}}$) refers to the vertical velocity in the layer of the model grid closest to the bottom.

## Vorticity budget analysis

We use the depth-integrated quasi-geostrophic vorticity balance equation[63].

$$\int_{-H}^{0}\left(\frac{\partial \zeta}{\partial t} + \mathbf{u} \cdot \nabla \zeta\right)dz = f_{\mathrm{o}}\left(w_{\mathrm{o}} - w_{-\mathrm{H}}\right) \qquad (3)$$

where $\mathbf{u}$ is the velocity of ocean current, $\zeta$ is the relative vorticity defined by $\zeta = \nabla \times \mathbf{u}$, $f_{\mathrm{o}}$ is the Coriolis parameter, $w_{\mathrm{o}}$ and $w_{-\mathrm{H}}$ correspond to the vertical velocity at the surface and the bottom grid of the model, respectively. The left-hand-side represents time tendency of vorticity and relative vorticity advection, while the right-hand-side represents the surface and bottom vortex stretching term.

## Data availability

The model code, input, and results are available at https://zenodo.org/records/6570222 provided by Hyogo[64]. They are also available at https://ecco.jpl.nasa.gov/drive/files/ECCO2/LLC1080_REG_AMS/Hyogo_et_al_2022.

## Code availability

The main codes used for generating results are accessible at https://zenodo.org/records/10812679.

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

## Acknowledgements

This study was supported by the Korea Institute of Marine Science & Technology Promotion (KIMST) under the funding provided by the Korean Ministry of Oceans and Fisheries (Grant No. RS-2023-00256677; PM24020) for T.P. and S.N. Additional support for T.P. was provided by the Korea Polar Research Institute (Grant No. PE24110). The Grants-in-Aid for Scientific Research (Grant No. 21K13989) from the Japanese Ministry of Education, Culture, Sports, Science, and Technology and the Inoue Science Research Award from the Inoue Science Foundation supported N.Y.

## Author contributions

T.P. and N.Y. designed the study and conducted the model data analysis. T.P., N.Y. and S.N. contributed to the interpretation of the results, development of the idea, and writing of the manuscript.

## Competing interests

The authors declare no competing interests.
