## [Peer Review File · Nature Communications]

Amundsen Sea Circulation Controls Bottom Upwelling and Antarctic Pine Island and Thwaites Ice Shelf MeltingREVIEWER COMMENTS

Reviewer #1 (Remarks to the Author):

This is a superb, timely, and well written paper that challenges the widely-held notion that ice-shelf melting in the Amundsen Sea region is primarily controlled by shelf-break westerly winds. The authors provide compelling evidence that this is probably incorrect, and that the more important mechanism are meanders on the Antarctic coastal current.

My only two (minor) criticisms of the paper are that

1) I think that the paper by Paolo and others, which in my opinion provides the most compelling evidence for a strong link -- not just during major events such as 2012 -- between ENSO and ice-shelf melting. This paper is based on satellite altimetry data and snowfall data that allow for a continuous time series of melt rates. I am not suggesting that the authors of the current paper need to go into detail on this, but they might mention that more work is needed to reconcile these different lines of evidence.

2) The authors are careful not to say too much about why there have been changes in the AAC that promote melting, and I recognize that more work is needed in the future. However, they do suggest that perhaps remote easterly winds are responsible. They mention this in the section titled "Roles of ocean surface stress on ocean circulation", but they do not mention this at all in the Discussion or the Abstract. I think that it would be helpful for readers to be reminded of this, even if it is not yet proven. It seems to me a very important point.

Reviewer #2 (Remarks to the Author):

Overall, this is an interesting study that provides some novel results. However, my major criticism is that the title and the summary provided in the abstract is not supported by the analysis. The title is misleading in a number of ways, and appears targeted at exaggerating the importance of the results to achieve publication in a high-impact journal. Therefore, I recommend a major revision, in which the authors re-frame the manuscript to more accurately reflect their main results.

My interpretation of the main result is that the ice-shelf melting in PIIS & Thwaites is strongly correlated to the barotropic (depth-averaged) circulation on the continental shelf, as captured by PC1. One component of this could be called the Antarctic Coastal Current, but the southward flow in Pine Island Trough is predominantly topographically-steered and not truly coastal. The circulation pattern is similar to that presented in several previous studies (including by some of these authors), and particularly similar to the flow of CDW around the shelf. There is no solid evidence (beyond a somewhat speculative and poorly reasoned correlation with wind stress) that the Antarctic Coastal Current outside of the Amundsen Sea is directly implicated in the melting of Pine Island or Thwaites ice shelves. Therefore, the title seems to be substantially overstating the novelty and significance of the results.

Abstract and key results:

Line 19-20: "Thermocline depth determines overall basal melt rates". This is presented as accepted fact here, but is quite a controversial statement, and the accuracy of the statement depends on the location: it is probably true that thermocline depth in front of Pine Island ice shelf strongly influences basal melt rates, while thermocline depth 100 km away may or may not be well correlated. Other factors, particularly circulation speed, also matter. Therefore, I don't think this statement can be used as written, but it could be rephrased to say that thermocline depth in Pine Island Bay is strongly

correlated to basal melt rates.

Line 23: The location of the AACC variability needs to be specified, unless the authors can demonstrate that this variability is coherent across large spatial scales. On reading the abstract, I assumed this meant the AACC in the Bellingshausen Sea. But I believe the authors are referring to the southward-flowing AACC in Pine Island Trough, which is potentially very misleading.

Line 21: it is the strength of the on-shelf circulation, that influences thermocline depth, not necessarily the coastal current. Indeed, it's not clear whether this mechanism relates to the seasonal variation in thermocline depth? Regardless, the notion that the circulation correlates strongly with thermocline depth is not a new one, and has been shown before. The novel result is that time-integrated vertical velocity from bottom Ekman convergence is quantitatively linked to the thermocline depth. This is a really interesting result and should have more prominence in the paper.

Line 24-26. What does this statement really mean? Is it truly a general result that can be widely applied? Surely it depends on the specific bathymetry in front of each ice shelf?

Line 26-28. The authors don't calculate the flow of mCDW onto the continental shelf (which is my interpretation of "on-shelf flow of mCDW" – but perhaps not what the authors mean, in which case the wording should be improved for clarity), only the strength of the on-shelf circulation. These two quantities may be related, but that isn't shown. The authors do show that the basal melting is primarily determined by the variability of this on-shelf circulation, which implies that either the two are related or the flow of mCDW onto the shelf is relatively unimportant. Regardless, I think it is difficult to conclude that the shelf-edge winds are unimportant from these results, since lagged or cumulative effects are not considered.

Did the authors attempt to correlate the cumulative zonal OSS at the shelf break with PC1? There is a clear theoretical argument (albeit not well expressed in the literature), for the zonal winds to correlate with on-shelf heat flux that would then accumulate. Therefore, one would not expect a strong instantaneous correlation between a flux and a quantity. Fig. 3e suggests there may be a lagged correlation, but it is hard to be sure.

Further specific comments:

Fig. 1 Please specify the shaded variable and units for each panel on the figure. Only having the variable in the caption makes it hard for the reader to interpret the figure.

In Fig. 1, c,d, what is the horizontal axis? It cannot be the same as e,f, since those panels are a zonal average. This makes the whole figure hard to interpret, as it appears that c,d and e,f are comparable sections.

Also in Fig. 1, what are the red and black lines in e,f? They don't seem to be in the caption.

In Fig. 2d, the time-integrated W_{EK} is shown, but barely discussed. Yet, to me there appears to be a strong anti-correlation here. It's not quite clear why, but this suggests a role for local surface stress in driving the variability seen.

Line 70-73: I find the use of the past tense here to be strange. I would prefer "hypothesize" and "use".

Line 84-86. This is in agreement with previous studies, which should be mentioned here.

Supplementary Figure 1 should be included in the paper, since this is a key component of the results.

Line 116-118: You don't show that the variability in the AACC is coherent across space and time.

Line 160-167: Is the proposed mechanism instantaneous? Surely there should be a lag between conditions in the Bellingshausen sea and those in the southern Amundsen Sea?

Reviewer #3 (Remarks to the Author):

Park et al use a numerical model to determine what causes changes in the ocean heat delivery toward ice shelves in West Antarctica. The topic is highly important, especially thinking in terms of future sea level rise, and therefore appropriate for Nature Communication. The mechanism proposed by the authors, that involves coastal current and bottom friction, is interesting and worth analysing it. However, my impression is that more analysis is required before confirming that this mechanism is at play in the model. Below I have some major comments that can help in this direction, plus some minor comments.

Major Comments

- Line 119-132: The proposed mechanism is interesting, but more analysis is required to support it. A full vorticity budget should be implemented here to understand the role of friction, buoyancy etc.
- Line 160-167 Interesting! Again, you could test this in your model looking at the evolution of the AACC between the Bellingshausen and Amundsen Sea, to see whether connectivity at interannual time scale can emerge. I would also refer to Flexas et al 2022 here ([10.1126/sciadv.abj9134](https://doi.org/10.1126/sciadv.abj9134)). Moreover, looking at changes in the AACC upstream and the Antarctic Slope Current in the eastern Amundsen Sea might help disentangle what drives variability in PIB (e.g. Silvano et al 2022; [10.1029/2022GL100646](https://doi.org/10.1029/2022GL100646)).
- Line 356-364: Here you apply a 13-month filter before applying EOF analysis. This implies that this type of analysis specifically looks at interannual and longer time scales. How does this affect the type of correlation you are calculating? How about lags? How about bottom friction? This requires further explanation.

Minor Comments

- Title: what do you mean by "control"? control on temporal variability, if so which time scale? If not, are you referring to steady state? This should be clarified starting from the title.
- Line 18: Ice shelf melting does not directly drive sea level rise.
- Line 26-28: also here, are you referring to steady state? Time variability? I would imagine that winds play a role at some time scale as well as in setting the time-mean currents. Please clarify.
- Line 62: recently it has been proposed time variability in the undercurrent, with baroclinic variability at decadal time scale that can affect ice shelf melting ([10.1029/2022GL100646](https://doi.org/10.1029/2022GL100646)).
- Line 335: what are the boundary conditions of the model?
- Line 336-347: has this model been validated before? If so, I would mention it.
- Figure 1: please add labels to the plots and specify what the red and black lines in panel e and f are. In panel c,d are you showing temperature anomalies, or average temperature during warm and cold periods? Please clarify.
- Figure s2b: please specify what black and grey line are.

Reviewer #1

We appreciate the reviewer's constructive feedback, which helped us improve our paper. We have revised it carefully following reviewer's suggestions. Details of these changes are in the next section.

This is a superb, timely, and well written paper that challenges the widely-held notion that ice-shelf melting in the Amundsen Sea region is primarily controlled by shelf-break westerly winds. The authors provide compelling evidence that this is probably incorrect, and that the more important mechanisms are meanders on the Antarctic coastal current.

My only two (minor) criticisms of the paper are that

1) I think that the paper by Paolo and others, which in my opinion provides the most compelling evidence for a strong link -- not just during major events such as 2012 -- between ENSO and ice-shelf melting. This paper is based on satellite altimetry data and snowfall data that allow for a continuous time series of melt rates. I am not suggesting that the authors of the current paper need to go into detail on this, but they might mention that more work is needed to reconcile these different lines of evidence.

We cited the study by Paolo et al. (2018):

<Lines 64-65> *"A strong link between ENSO and ice shelf melting are examined focusing on the variations in both ice shelf thickness and mass³⁶."*

2) The authors are careful not to say too much about why there have been changes in the AAAC that promote melting, and I recognize that more work is needed in the future. However, they do suggest that perhaps remote easterly winds are responsible. They mention this in the section titled "Roles of ocean surface stress on ocean circulation", but they do not mention this at all in the Discussion or the Abstract. I think that it would be helpful for readers to be reminded of this, even if it is not yet proven. It seems to me a very important point.

We have included analysis results and discussion regarding the AACC:

<Lines 25-27: Abstract> *"We suggest that a strong or weak ocean circulation possibly linked to remote winds in the Bellingshausen Sea generates a correspondingly strong or weak bottom Ekman convergence, which modulates mCDW upwelling and TD variability."*

<Lines 183-188: Results> *"We investigated the evolution of the AACC between the Bellingshausen and Amundsen Seas and identified a strong and consistent connection between the on-shelf circulation in the eastern Amundsen Sea and various locations along the path of the AACC (Supplementary Figure S6). This finding confirms that the AACC influences the ocean circulation in the eastern Amundsen Sea downstream instantaneously, mainly through its westward transport pathway^{40,45}."*

<Lines 228-230: Discussion and implications> *"Further studies on the upstream influences of the AACC and the wind forcings in the Bellingshausen Sea can help to elucidate the mechanism for the melting of Antarctic ice shelves."*

Reviewer #2

We appreciate the reviewer's constructive feedback, which helped us improve our paper. We have revised it carefully following reviewer's suggestions. Details of these changes are in the next section.

Overall, this is an interesting study that provides some novel results. However, my major criticism is that the title and the summary provided in the abstract is not supported by the analysis. The title is misleading in a number of ways, and appears targeted at exaggerating the importance of the results to achieve publication in a high-impact journal. Therefore, I recommend a major revision, in which the authors re-frame the manuscript to more accurately reflect their main results.

My interpretation of the main result is that the ice-shelf melting in PIIS & Thwaites is strongly correlated to the barotropic (depth-averaged) circulation on the continental shelf, as captured by PC1. One component of this could be called the Antarctic Coastal Current, but the southward flow in Pine Island Trough is predominantly topographically-steered and not truly coastal. The circulation pattern is similar to that presented in several previous studies (including by some of these authors), and particularly similar to the flow of CDW around the shelf. There is no solid evidence (beyond a somewhat speculative and poorly reasoned correlation with wind stress) that the Antarctic Coastal Current outside of the Amundsen Sea is directly implicated in the melting of Pine Island or Thwaites ice shelves. Therefore, the title seems to be substantially overstating the novelty and significance of the results.

We have revised the title and abstract to more accurately reflect the key findings of our study, ensuring they are not overstated. We agree that while the flow in Pine Island Trough is predominantly topographically-steered, our classification of the Antarctic Coastal Current as part of the on-shelf circulation was intended to include broader circulation influences on the ice shelf melting. We have made this clearer and added support for it. Furthermore, the revised manuscript includes additional evidence about the influence of the Antarctic Coastal Current beyond the Amundsen Sea, including the Bellingshausen Sea.

Abstract and key results:

Line 19-20: "Thermocline depth determines overall basal melt rates". This is presented as accepted fact here, but is quite a controversial statement, and the accuracy of the statement depends on the location: it is probably true that thermocline depth in front of pine island ice shelf strongly influences basal melt rates, while thermocline depth 100 km away may or may not be well correlated. Other factors, particularly circulation speed, also matter. Therefore, I don't think this statement can be used as written, but it could be rephrased to say that thermocline depth in Pine Island Bay is strongly correlated to basal melt rates.

To investigate the spatial pattern of thermocline depth and its relation to melt rates, we conducted correlation analysis between the thermocline depth and the melt rates of both Pine Island and Thwaites Ice Shelves (Figure R1). This analysis revealed a stronger correlation within the Pine Island Bay, suggesting thermocline depth in the Pine Island Bay can strongly

influence the basal melt rates. The abstract has been rewritten and an explanation for the Figure R1 has been included as follows:

<Lines 19-20>

“...modified Circumpolar Deep Water (mCDW), determines the overall basal melt rates...”

→ “...modified Circumpolar Deep Water (mCDW), at the PIIS/TIS front is strongly correlated to the basal melt rates...”

<Lines 92-93>

“Our results demonstrate a strong correlation between the TD and basal melt rates (Supplementary Fig. S1)...”

→ “Our results demonstrate a strong correlation between the TD in the PIB and basal melt rates (Supplementary Fig. S1)...”

Figure R1 (Supplementary Figure S1 in the manuscript): Correlation coefficients between 0.5°C isotherm depth and the melt rates of both Pine Island and Thwaites Ice Shelves. Statistically insignificant regions in 95%-level are shaded transparently. Black color-coded contours indicate the bathymetry of 500 m and 700 m depth.

Line 23: The location of the AACC variability needs to be specified, unless the authors can demonstrate that this variability is coherent across large spatial scales. On reading the abstract, I assumed this meant the AACC in the Bellingshausen Sea. But I believe the authors are referring to the southward-flowing AACC in Pine Island Trough, which is potentially very misleading.

Following the reviewer’s general comments above and specified comments made on Line 21 below, we revised the term 'AACC' to denote 'on-shelf circulation in the eastern Amundsen Sea' throughout the manuscript to prevent any misunderstanding with the term used for coastal current.

Line 21: it is the strength of the on-shelf circulation, that influences thermocline depth, not necessarily the coastal current. Indeed, it's not clear whether this mechanism relates to the seasonal variation in thermocline depth? Regardless, the notion that the circulation correlates strongly with thermocline depth is not a new one, and has been shown before. The novel result is that time-integrated vertical velocity from bottom Ekman convergence is quantitatively linked to the thermocline depth. This is a really interesting result and should have more prominence in the paper.

Regarding the remarks, “it is the strength of the on-shelf circulation, that influences thermocline depth, not necessarily the coastal current”, we have revised the term 'AACC' to denote 'on-shelf circulation in the eastern Amundsen Sea' throughout the manuscript as per our previous response.

Regarding the comment on the seasonality of thermocline depth, we conducted further analysis using data from each month (Figure R2):

<Lines 123-128> *“We revisited our analysis by using monthly data to investigate the seasonal variability and compared this with the identified interannual variability (Supplementary Fig. S2). Our findings indicate that ocean circulation showed pronounced seasonal variability, while the thermocline depth does not vary much seasonally. Consequently, the response of the thermocline depth to the ocean circulation is more pronounced in the longer than seasonal timescale.”*

Figure R2 (Supplementary Figure S2 in the manuscript): The leading mode's principal component (PC1) of ocean current (black line), southward AACC transport across 74.5°N, 104-102°W in X-X' in the Pine Island Bay (gray dotted line), and 0.5°C isotherm depth (red line). **a**, All variables are derived from the monthly data. **b**, Seasonal variations of all variables are removed using a 13-month moving average filter.

We appreciate the recognition of our findings regarding the time-integrated vertical velocity from bottom Ekman convergence and its link to thermocline depth. We changed the title and updated descriptions to emphasize bottom-driven upwelling process:

<Title> *“Antarctic Coastal Current Controls Thwaites and Pine Island Ice Shelf Melting in the Amundsen Sea, West Antarctica”*

→ *“Amundsen Sea Circulation Generates Bottom-Driven Upwelling and Controls Pine Island and Thwaites Ice Shelf Melting, West Antarctica”*

<Lines 129-134> *“we suggest the horizontal divergence at the bottom Ekman layer is responsible for the Ekman pumping of warm mCDW into the ocean interior; which, in turn,*

affects the displacement of the TD. It was confirmed because time-integrated vertical velocity fluctuates similarly to the TD (Fig. 2c). This analysis suggests that stronger cyclonic ocean circulation may enhance mCDW convergence, as indicated by the larger negative relative vorticity (see Methods)."

Line 24-26. What does this statement really mean? Is it truly a general result that can be widely applied? Surely it depends on the specific bathymetry in front of each ice shelf?

We rewrote the sentence:

<Lines 23-25>

"The convergence occurs because the AACC paths meander along the intricate bathymetry of the Antarctic coast."

→ *"The TD variability occurs because the on-shelf circulation meanders following the submarine glacial trough creating vertical velocity through bottom Ekman dynamics."*

Line 26-28. The authors don't calculate the flow of mCDW onto the continental shelf (which is my interpretation of "on-shelf flow of mCDW" – but perhaps not what the authors mean, in which case the wording should be improved for clarity), only the strength of the on-shelf circulation. These two quantities may be related, but that isn't shown. The authors do show that the basal melting is primarily determined by the variability of this on-shelf circulation, which implies that either the two are related or the flow of mCDW onto the shelf is relatively unimportant. Regardless, I think it is difficult to conclude that the shelf-edge winds are unimportant from these results, since lagged or cumulative effects are not considered.

Did the authors attempt to correlate the cumulative zonal OSS at the shelf break with PC1? There is a clear theoretical argument (albeit not well expressed in the literature), for the zonal winds to correlate with on-shelf heat flux that would then accumulate. Therefore, one would not expect a strong instantaneous correlation between a flux and a quantity. Fig. 3e suggests there may be a lagged correlation, but it is hard to be sure.

Regarding the flow of the mCDW, we rephrased the sentence for clarity:

<Line 29>

"the on-shelf flow of mCDW,"

→ *"the inflow of mCDW onto the shelf,"*

We compared the cumulative zonal OSS at the shelf break and the PC1 (Figure R3a):

<Lines 198-203> *"We assessed the time-integrated ocean surface stress at the shelf break using an approach similar to that of previous studies^{23,44} and compared it with the ocean circulation on the shelf by PC timeseries (Supplementary Fig. S7a). Our analysis does not support instantaneous response either; the response of the PC does not coincide directly with the wind forcing. The wind forcing precedes the ocean circulation by two years."*

Figure R3 (Supplementary Figure S7 in the manuscript): The leading mode's principal component (PC1) of ocean current (black line) with **a**, time-integrated zonal ocean surface stress (red line) and **b**, zonal ocean surface stress after applying 5-year moving average filter (red line). The ocean surface stress is averaged on the continental shelf break within the box A in Fig. 4c in the main text.

To further validate our findings in accordance with previous studies regarding the shelf-edge winds, we conducted additional analysis employing a 5-year filter to emphasize the decadal timescale:

<Lines 203-209>

“To investigate decadal ocean response to winds, we further conducted our analysis using a 5-year moving average filter (Supplementary Fig. S7b), consistent with a prior study⁴³. We found a decadal-scale negative correlation between the westerly winds and on-shelf ocean circulation, aligning with a previous study⁴³ that linked an easterly wind anomaly to an eastward undercurrent that can promote the heat transport onto the shelf. However, our model output is not long enough to evaluate the robustness of this mechanism.”

We emphasized the significance of winds as having a local impact within the PIB and a remote influence in the Bellingshausen Sea, rather than focusing on the shelf-edge winds:

<Lines 169-172>

“Conversely, in this study, local and remote wind forcings over the continental shelf are proposed to be linked with increased ice shelf melting (Fig. 4a,d).

Locally in the eastern Amundsen Sea, our investigation on the OSS (see Methods) (Fig. 4d,f),...”

<Lines 216-222>

“We suggest a new perspective on the role of winds, indicating that the northward local winds in the PIB, along with the remote easterly winds over the continental shelf in the Bellingshausen Sea, are crucial factors in strengthening the circulation in the eastern Amundsen Sea. We did not find a significant relationship between westerly winds off the continental shelf break and ice shelf melt rates within the PIB on the interannual timescale, despite prior studies^{23,43,44} suggesting that zonal winds at the shelf break could influence ice shelf melting.”

Further specific comments:

Fig. 1 Please specify the shaded variable and units for each panel on the figure. Only having the variable in the caption makes it hard for the reader to interpret the figure.

We modified Figure 1.

In Fig. 1, c,d, what is the horizontal axis? It cannot be the same as e,f, since those panels are a zonal average. This makes the whole figure hard to interpret, as it appears that c,d and e,f are comparable sections.

We modified Figure 1.

Also in Fig. 1, what are the red and black lines in e,f? They don't seem to be in the caption.

We revised the figure caption:

<Figure 1> *“The black contour line represents 0.5°C isothermal line averaged for the total model simulation, while the red contour line shows 0.5°C isothermal line averaged in high- and low-melt years in each figure.”*

In Fig. 2d, the time-integrated W_EK is shown, but barely discussed. Yet, to me there appears to be a strong anti-correlation here. It's not quite clear why, but this suggests a role for local surface stress in driving the variability seen.

We have provided a discussion for Fig. 2d:

<Lines 137-144> *“The magnitude of the TD variability estimated from the surface Ekman upwelling (Fig. 2d) is about 70% smaller than that estimated from near-bottom vertical velocity (Fig. 2c). Furthermore, the surface Ekman pumping does not exhibit coherent variations with the TD variability; occasionally, it presents an opposing phase to the TD variability (Fig. 2d). This underlines the minor role of surface Ekman pumping in influencing the TD variability, which is consistent with results of previous studies^{32,43}. These studies have shown that surface Ekman pumping is unlikely to be the main driver of influencing the variability in CDW layer thickness in the Amundsen Sea on interannual or decadal timescale.”*

Line 70-73: I find the use of the past tense here to be strange. I would prefer “hypothesize” and “use”.

<Lines 73-74> We have made the suggested change.

Line 84-86. This is in agreement with previous studies, which should be mentioned here.

We revised this:

<Lines 90-91> *“This corresponds to isotherm fluctuations with similar amplitude observed over five years in a previous observational study²⁴”*

Supplementary Figure 1 should be included in the paper, since this is a key component of the results.

We agree. We moved the Supplementary Figure 1 to Figure 3 in the main text.

Line 116-118: You don't show that the variability in the AACC is coherent across space and time.

We calculated the correlation between the on-shelf circulation and the AACC transport and we suggest the AACC is coherent over wide area in the eastern Amundsen Sea and the western Bellingshausen Sea without time-delay within a year (Figure R4):

<Lines 183-186>

“We investigated the evolution of the AACC between the Bellingshausen and Amundsen Seas and identified a strong and consistent connection between the on-shelf circulation in the eastern Amundsen Sea and various locations along the path of the AACC (Supplementary Figure S6).”

Figure R4 (Supplementary Figure S6 in the revised manuscript): **a**, Regression coefficients (arrows) of ocean current anomalies averaged above the 0.5°C isotherm depth on the leading mode's principal component (PC1) of ocean current superimposed with their correlation coefficients (color shading) of the PC1 and zonal ocean current. Statistically insignificant regions in 90%-level are shaded transparently. Three sections (green line) are for the lead-lag correlation plots in **b,c,d**. **b,c,d**, Lead-lag correlations (black line) between PC1 and volume transports each section at a 95% confidence level (red dotted line).

Line 160-167: Is the proposed mechanism instantaneous? Surely there should be a lag between conditions in the Bellingshausen sea and those in the southern Amundsen Sea?

We looked for the time lags on the relationship between wind patterns and ocean circulation, but could not find lagged responses (Figure R5):

<Lines 180-183> “Although a time lag may exist on a monthly scale, our analysis with 13-months running averages suggests that the response is near instantaneous and we did not reveal any time lag between the winds in the Bellingshausen Sea and the ocean circulation in the eastern Amundsen Sea (Supplementary Fig. S5).”

Figure R5 (Supplementary Figure S5 in the manuscript): Lagged regression coefficients (arrows) of wind anomalies on the leading mode's principal component (PC1) of ocean current in the eastern Amundsen Sea, superimposed with their correlation coefficients (color shading) of the zonal wind and the PC1. Statistically insignificant regions in the 90%-level are transparently shaded. The "Lag (month)" denotes a time delay of the ocean circulation in the eastern Amundsen Sea to wind patterns in month. A green color-coded contour indicates the bathymetry of 700 m and 1000 m depth.

Reviewer #3

We appreciate the reviewer's constructive feedback, which helped us improve our paper. We have revised it carefully following reviewer's suggestions. Details of these changes are in the next section.

Park et al use a numerical model to determine what causes changes in the ocean heat delivery toward ice shelves in West Antarctica. The topic is highly important, especially thinking in terms of future sea level rise, and therefore appropriate for Nature Communication. The mechanism proposed by the authors, that involves coastal current and bottom friction, is interesting and worth analysing it. However, my impression is that more analysis is required before confirming that this mechanism is at play in the model. Below I have some major comments that can help in this direction, plus some minor comments.

Major Comments

- Line 119-132: The proposed mechanism is interesting, but more analysis is required to support it. A full vorticity budget should be implemented here to understand the role of friction, buoyancy etc.

We conducted a vorticity budget analysis, the details of which are provided in the Methods section, and the results are presented as follows (Figure R1):

<Lines 145-154> *“We performed an analysis of the depth-integrated vorticity budget that takes into account the tendency of vorticity, vorticity advection, along with the planetary vorticity due to vertical velocities at both the surface and bottom (see Methods) (Supplementary Fig. S3). The most prominent factors were the advection of relative vorticity and the planetary vorticity due to vertical velocity at the bottom. We noted that the tendency of vorticity is negligibly small, which can become important in shorter timescales, e.g., super-inertial scales such as daily or shorter, but not over monthly or longer timescales relevant to this study. The residual term, which includes effects of horizontal mixing and computational errors, contributes to a minor 3% of the vorticity advection. Notably, the planetary vorticity due to bottom vertical velocity primarily acts in response to the advection of relative vorticity.”*

Figure R1 (Supplementary Figure S3 in the manuscript): **a**, Depth-integrated vorticity budget terms averaged south of X-X' in Fig. 1a in the main text. **b**, Seasonal variations of all the budget terms are removed using a 13-month moving average filter.

- Line 160-167 Interesting! Again, you could test this in your model looking at the evolution of the AACC between the Bellingshausen and Amundsen Sea, to see whether connectivity at interannual time scale can emerge. I would also refer to Flexas et al 2022 here (10.1126/sciadv.abj9134). Moreover, looking at changes in the AACC upstream and the Antarctic Slope Current in the eastern Amundsen Sea might help disentangle what drives variability in PIB (e.g. Silvano et al 2022; 10.1029/2022GL100646).

<Lines 183-186>

“We investigated the evolution of the AACC between the Bellingshausen and Amundsen Seas and identified a strong and consistent connection between the on-shelf circulation in the eastern Amundsen Sea and various locations along the path of the AACC (Supplementary Figure S6).”

Figure R2 (Supplementary Figure S6 in the revised manuscript): a, Regression coefficients (arrows) of ocean current anomalies averaged above the 0.5°C isotherm depth on the leading mode’s principal component (PC1) of ocean current superimposed with their correlation coefficients (color shading) of the PC1 and zonal ocean current. Statistically insignificant regions in 90%-level are shaded transparently. Three sections (green line) are for the lead-lag correlation plots in b,c,d. b,c,d, Lead-lag correlations (black line) between PC1 and volume transports each section at a 95% confidence level (red dotted line).

Following the suggestion, we have included the study by Flexas et al. (2022) where we discuss previous studies related to the Antarctic Coastal Current:

<Lines 177-180>

“This is consistent with previous studies showing that the prevailing easterly winds were examined as a driving force for AACC variability, along with buoyancy forces from meltwater and runoff in the Bellingshausen Sea and the West Antarctic Peninsula continental shelf⁴⁶⁻⁴⁹.”

Regarding the reviewer’s suggestion on the Antarctic Slope Current, although we did not examine the Antarctic Slope Current in detail; rather, we focused on the on-shelf ocean circulation dynamics, we conducted similar analysis to the study by Silvano et al. (2022) suggested by the reviewer and explained the results on a decadal timescale:

<Lines 203-209>

“To investigate decadal ocean response to winds, we further conducted our analysis using a 5-year moving average filter (Supplementary Fig. S7b), consistent with a prior study⁴³. We found a decadal-scale negative correlation between the westerly winds and on-shelf ocean circulation, aligning with a previous study⁴³ that linked an easterly wind anomaly to an eastward undercurrent that can promote the heat transport onto the shelf. However, our model output is not long enough to evaluate the robustness of this mechanism.”

• Line 356-364: Here you apply a 13-month filter before applying EOF analysis. This implies that this type of analysis specifically looks at interannual and longer time scales. How does this affect the type of correlation you are calculating? How about lags? How about bottom friction? This requires further explanation.

We conducted our analysis again using data from each month and compared the timeseries using monthly data and low-pass filtered data (Figure R3):

<Lines 123-128>

“We revisited our analysis by using monthly data to investigate the seasonal variability and compared this with the identified interannual variability (Supplementary Fig. S2). Our findings indicate that ocean circulation showed pronounced seasonal variability, while the thermocline depth does not vary much seasonally. Consequently, the response of the thermocline depth to the ocean circulation is more pronounced in the longer than seasonal timescale.”

*Figure R3 (Supplementary Figure S2 in the manuscript): The leading mode’s principal component (PC1) of ocean current (black line), southward AACC transport across 74.5°N, 104-102°W in X-X’ in the Pine Island Bay (gray dotted line), and 0.5°C isotherm depth (red line). **a**, All variables are derived from the monthly data. **b**, Seasonal variations of all variables are removed using a 13-month moving average filter.*

<Lines 421-424>

“The use of a 13-month moving average filter serves to remove the effects of shorter-term fluctuations, such as those associated with seasonal variability. Consequently, the correlations identify relationships over interannual to decadal timescales.”

Although not included in the manuscript, following the reviewer’s suggestion, we assessed the lead-lag correlation coefficients between the PC1 and variables including the AACC transport, bottom vertical velocity, and the depth of the 0.5°C isotherm (Figure R4). To investigate the potential prolonged effects, we integrated the bottom vertical velocity over time. This analysis revealed no significant lag between ocean circulation patterns and both the depth of the 0.5°C isotherm and the time-integrated bottom vertical velocity on timescales longer than one year, suggesting that the oceanic response occurs within a year.

Figure R4: Lead-lag correlations between PC1 and both **a**, bottom vertical velocity and **b**, 0.5°C isotherm depth. Seasonal variations of all variables are removed using a 13-month moving average filter.

Minor Comments

- Title: what do you mean by “control”? control on temporal variability, if so which time scale? If not, are you referring to steady state? This should be clarified starting from the title.

We revised the title following your and other reviewer’s comments. Throughout the manuscript, we have distinctly emphasized the interannual variability that we are focusing on; yet, we didn’t specify this in the title in order to give more highlight to the underlying mechanism:

<Title>

“Antarctic Coastal Current Controls Thwaites and Pine Island Ice Shelf Melting in the Amundsen Sea, West Antarctica”

→ *“Amundsen Sea Circulation Generates Bottom-Driven Upwelling and Controls Pine Island and Thwaites Ice Shelf Melting, West Antarctica”*

- Line 18: Ice shelf melting does not directly drive sea level rise.

We rephrased this sentence:

<Line 18>

“raising global sea levels.”

→ *“impacting global sea levels.”*

- Line 26-28: also here, are you referring to steady state? Time variability? I would imagine that winds play a role at some time scale as well as in setting the time-mean currents. Please clarify.

We refer to time variability:

<Line 28>

“...the off-shelf zonal winds primarily controls...”

→ “...the interannual variability of off-shelf zonal winds primarily influences...”

In response to this comment and the following subsequent comment regarding Line 62, we did the same analysis as Silvato et al. (2022) conducted (Figure R5):

<Lines 198-209>

“We assessed the time-integrated ocean surface stress at the shelf break using an approach similar to that of previous studies^{23,44} and compared it with the ocean circulation on the shelf by PC timeseries (Supplementary Fig. S7a). Our analysis does not support instantaneous response either; the response of the PC does not coincide directly with the wind forcing. The wind forcing precedes the ocean circulation by two years. To investigate decadal ocean response to winds, we further conducted our analysis using a 5-year moving average filter (Supplementary Fig. S7b), consistent with a prior study⁴³. We found a decadal-scale negative correlation between the westerly winds and on-shelf ocean circulation, aligning with a previous study⁴³ that linked an easterly wind anomaly to an eastward undercurrent that can promote the heat transport onto the shelf. However, our model output is not long enough to evaluate the robustness of this mechanism.”

*Figure R5 (Supplementary Figure S7 in the manuscript): The leading mode's principal component (PC1) of ocean current (black line) with **a**, time-integrated zonal ocean surface stress (red line) and **b**, zonal ocean surface stress after applying 5-year moving average filter (red line). The ocean surface stress is averaged on the continental shelf break within the box A in Fig. 4c in the main text.*

- Line 62: recently it has been proposed time variability in the undercurrent, with baroclinic variability at decadal time scale that can affect ice shelf melting (10.1029/2022GL100646).

Please refer to the response provided above.

- Line 335: what are the boundary conditions of the model?

<Lines 391-392> *“Surface and lateral forcing for the 1992–2017 period”*

- Line 336-347: has this model been validated before? If so, I would mention it.

<Lines 394-396> *“This model and its further downscaled version have been used and evaluated for off-shelf and on-shelf hydrography and circulation in the Amundsen Sea and Bellingshausen Sea^{22,39,58-60}.”*

- Figure 1: please add labels to the plots and specify what the red and black lines in panel e and f are. In panel c,d are you showing temperature anomalies, or average temperature during warm and cold periods? Please clarify.

We have revised Figure 1.

- Figure s2b: please specify what black and grey line are.

We have revised Figure 1.

REVIEWERS' COMMENTS

Reviewer #2 (Remarks to the Author):

The authors have done a thorough job of responding to my comments (and those of the other reviewers). I am satisfied that the paper is now worthy of publication with some minor changes outlined below.

Overall, I still have two scientific questions:

1. What sets the long timescale for the response of TD to the ocean current? Why is there seasonal variability in the currents that is not present in the TD?
2. Why is there (near) zero lag between the conditions in the Bellingshausen Sea and the southern Amundsen Sea?

I recognise that the authors may be unable to answer these questions. If they can, that is ideal, and would strengthen the paper. However, at a minimum, I ask that these issues are acknowledged in the discussion and some plausible explanations or avenues for further study are discussed.

Minor changes:

L21: remains -> remain

L24: add comma between "trough" and "creating"

L25-26: add commas around "possibly linked to remote winds in the Bellingshausen Sea"

L28: "primarily influences" is ambiguous. Change to "is the primary driver of"?

L131: "It was confirmed" sounds odd. How about "This hypothesis is supported by the fact that..."

L148 and 153: "planetary vorticity due to vertical velocity at the bottom". This is unclear – vertical velocity does not create planetary vorticity! Do you mean vortex stretching acting on planetary vorticity?

L201: Here the "instantaneous response" is in response to the integrated forcing? If so, that is rather confusing, since by definition a response to the time-integrated forcing is not instantaneous. I suggest rephrasing.

Reviewer #3 (Remarks to the Author):

I found the manuscript strongly improved. My last comment is whether surface buoyancy forcing (local and remote) could affect basal melt and thermocline depth in the Eastern Amundsen Sea. Few words on this might help further strengthen the conclusions of the manuscript.

Reviewer #2 (Remarks to the Author):

The authors have done a thorough job of responding to my comments (and those of the other reviewers). I am satisfied that the paper is now worthy of publication with some minor changes outlined below.

Overall, I still have two scientific questions:

1. What sets the long timescale for the response of TD to the ocean current? Why is there seasonal variability in the currents that is not present in the TD?

We further compared the TD with monthly and time-integrated vertical velocity in the bottom layer (Figure R1c).

<Line 125-129>

“We found that the pronounced seasonal variability in ocean currents does not manifest similarly in the TD because response of the TD is moderated by the slower and cumulative changes in vertical velocity in the bottom layer, which filters out short-term fluctuations, such as those that occur during seasonal periods (Supplementary Fig. 2c).”

Figure R1 (Supplementary Figure 2 in the manuscript). The leading mode's principal component (PC1)

of ocean current (black line), southward Antarctic Coastal Current (AACC) transport across 74.4°S, 104–101.5°W in Pine Island Bay (grey dotted line), and 0.5°C isotherm depth (red line). **a** All variables are derived from the monthly data. **b** The same as in **a** but seasonal variations of all variables are removed using a 13-month moving average filter. **c** A comparison of the monthly vertical velocity anomaly in the bottom layer (black line) and its time integration (red line). This time-integrated vertical velocity does not differ much from the one with an additional 13-month moving average filter in the main text.

2. Why is there (near) zero lag between the conditions in the Bellingshausen Sea and the southern Amundsen Sea?

<Line 194-198>

“The West Antarctic continental shelf is characterized by strong circumpolar winds that encircle the continent. These winds can generate coastal-trapped Kelvin waves that rapidly transmit barotropic current anomalies westward along the Antarctic coast within hours to days^{50,51}. This atmospheric connectivity can lead to simultaneous weather patterns across these regions, contributing to the near-zero lag in conditions.”

I recognise that the authors may be unable to answer these questions. If they can, that is ideal, and would strengthen the paper. However, at a minimum, I ask that these issues are acknowledged in the discussion and some plausible explanations or avenues for further study are discussed.

Minor changes:

L21: remains -> remain

Done

L24: add comma between “trough” and “creating”

Done

L25-26: add commas around “possibly linked to remote winds in the Bellingshausen Sea”

Done

L28: “primarily influences” is ambiguous. Change to “is the primary driver of”?

Done

L131: “It was confirmed” sounds odd. How about “This hypothesis is supported by the fact that...”

Done

L148 and 153: “planetary vorticity due to vertical velocity at the bottom”. This is unclear – vertical velocity does not create planetary vorticity! Do you mean vortex stretching acting on planetary vorticity?

<Line 138-147>

We replaced “planetary vorticity due to vertical velocity” with “vortex stretching term due to vertical velocity”.

L201: Here the “instantaneous response” is in response to the integrated forcing? If so, that is rather confusing, since by definition a response to the time-integrated forcing is not instantaneous. I suggest rephrasing.

We rephrased the sentence.

<Line 211-212>

“Our analysis does not support instantaneous response either; the response of the PC does not coincide directly with the wind forcing. The wind forcing precedes the ocean circulation by two years.”

→ “The PC response, even after accounting for time-integration of the winds, does not coincide directly with the wind forcing; rather, the winds precede by two years the ocean circulation.”

Reviewer #3 (Remarks to the Author):

I found the manuscript strongly improved. My last comment is whether surface buoyancy forcing (local and remote) could affect basal melt and thermocline depth in the Eastern Amundsen Sea. Few words on this might help further strengthen the conclusions of the manuscript.

<Line 160-167>

“A number of prior studies have investigated the influence of buoyancy forcing on the melting of ice shelves. Several studies have proposed that sea ice formation and cooling due to local air-sea heat exchange within the polynyas in the PIB can drive seasonal fluctuations in the TD^{24,38}. Meanwhile, there are studies suggesting that the local surface buoyancy forcing is unlikely to be the main driver of decadal variability of the ice shelf melting in the eastern Amundsen Sea^{26,43}. While we mainly investigated interannual to decadal timescale variability, we cannot rule out that surface buoyancy forcing could also affect variability shorter than those timescales.”